# Three-Dimensional Models of the Dental Pulp: Bridging Fundamental Biology and Regenerative Therapy

**DOI:** 10.3390/ijms262210960

**Published:** 2025-11-12

**Authors:** Rana Smaida, Guoqiang Hua, Nadia Benkirane-Jessel, Florence Fioretti

**Affiliations:** 1INSERM (National Institute of Health and Medical Research) UMR1260 Regenerative Nanomedicine, University of Strasbourg, 1 Rue Eugène Boeckel, 67081 Strasbourg, France; rana.smaida@lamina.one (R.S.); g.hua@unistra.fr (G.H.); nadia.jessel@inserm.fr (N.B.-J.); 2Lamina Therapeutics, 1 Rue Eugène Boeckel, 67000 Strasbourg, France; 3Faculté de Chirurgie Dentaire de Strasbourg, University of Strasbourg, 8 Rue Sainte-Elisabeth, 67000 Strasbourg, France; 4Pôle de Médecine et Chirurgie Bucco-Dentaire, Hôpitaux Universitaires de Strasbourg, 1 Place de l’Hôpital, 67000 Strasbourg, France

**Keywords:** dental pulp, 3D culture, organoid, spheroid, regenerative endodontics, biomaterials, vascularization, tooth-on-a-chip, tissue engineering, dental stem cells

## Abstract

The dental pulp is a dynamic connective tissue essential for tooth vitality, sensory function, immune defense, and reparative dentinogenesis. Conventional endodontic procedures, while effective in eradicating infection, often result in a non-functional, devitalized tooth, highlighting the need for biologically based regenerative approaches. The emergence of three-dimensional (3D) culture systems has transformed pulp biology and endodontic research by providing physiologically relevant microenvironments that better reproduce the dentino-pulp interface, vascular and neural networks, and immune interactions. This review synthesizes current advances in 3D dental pulp modeling, from scaffold-based and hydrogel systems to spheroids, organoids, bioprinted constructs, and microfluidic “tooth-on-a-chip” platforms. Each system’s composition, biological relevance, and translational potential are critically examined with respect to odontogenic differentiation, angiogenesis, neurogenesis, and inflammatory response. Applications in disease modeling, biomaterial screening, and regenerative endodontics are highlighted, showing how these models bridge fundamental biology and therapeutic innovation. Finally, we discuss key challenges including vascularization, innervation, standardization, and clinical translation, and propose integrative strategies combining bioprinting, stem-cell engineering, and organ-on-chip technologies to achieve functional pulp regeneration. Overall, 3D pulp models represent a paradigm shift from reductionist cultures to bioinstructive, patient-relevant platforms that accelerate the development of next-generation endodontic therapies.

## 1. Introduction

The dental pulp is a specialized connective tissue that plays a central role in tooth vitality, sensory perception, immune defense, and reparative dentin formation. Its unique composition, which includes odontoblasts, fibroblasts, mesenchymal stem cells, immune cells, and a complex vascular and neural network allows the pulp to respond dynamically to injury, infection, and mechanical stress. When the pulp is damaged or devitalized, as in trauma or conventional endodontic treatment, these essential functions are lost, leaving the tooth non-responsive and more susceptible to reinfection. Conventional endodontic procedures, although effective in removing infected tissue, fail to restore the physiological architecture and function of the pulp. This limitation has driven the search for regenerative strategies capable of restoring functional pulp tissue. However, the complex biology of the dentin-pulp complex, including the diversity of cell types, the intricate interplay of signaling pathways, and the confined anatomical space, poses significant challenges for translational research [1,2,3].

To make meaningful progress in pulp regeneration, the development of appropriate preclinical in vitro models is essential. Three-dimensional (3D) culture systems represent an intermediate step between traditional two-dimensional (2D) monolayer cultures and animal models, as they better reproduce the native architecture, oxygen, and nutrient gradients, and multicellular interactions of the tissue. While 2D cultures have provided valuable insights into stem-cell behavior, odontoblast differentiation, and material biocompatibility, they cannot fully recapitulate the complex microenvironment of the pulp. Animal models, although physiologically relevant, are costly, low throughput, and subject to interspecies differences that may limit translational applicability [4,5].

Three-dimensional culture systems have therefore emerged as powerful tools that more faithfully recapitulate the native pulp microenvironment. By supporting realistic cell–cell and cell–matrix interactions, promoting odontogenic differentiation, and enabling the formation of vascular and neural networks, 3D models allow for the study of both physiological and pathological pulp states. These models are increasingly used to analyze dentin–pulp interactions, vascularization, neurogenesis, inflammatory responses, and the effects of restorative materials and bioactive molecules. Advances in biomaterials, hydrogels, scaffold fabrication, 3D bioprinting, and microfluidic technologies have enabled the creation of increasingly complex pulp constructs. Organoid systems, prevascularized spheroids, scaffold-free cell aggregates, and tooth-on-a-chip platforms now allow mechanistic studies, drug, and biomaterial testing, and disease modeling in ways that were previously impossible [6].

By integrating multiple cell types, including dental pulp stem cells (DPSCs), stem cells from the apical papilla (SCAPs), periodontal ligament stem cells (PDLSCs), endothelial cells, and immune components, these 3D systems can reproduce the cellular heterogeneity and intercellular crosstalk of native pulp tissue. Microfluidic and organ-on-chip technologies provide dynamic perfusion, nutrient exchange, and mechanical stimuli that mimic physiological conditions, while scaffold and hydrogel innovations allow precise spatial control, biofactor delivery, and prevascularization. These advancements make it possible to evaluate biocompatibility, odontogenic potential, angiogenesis, neurogenesis, and inflammatory responses under controlled, physiologically relevant conditions [7,8,9].

This review surveys current 3D dental pulp models, their design strategies, advantages, and limitations, as well as their applications in physiological and pathological contexts and regenerative therapy. It also highlights emerging strategies to enhance vascularization, innervation, multicellular integration, and clinical translation, emphasizing the transformative potential of 3D models for the next generation of regenerative endodontic therapies. These bioengineered constructs bridge the gap between in vitro experimentation and in vivo function, providing an indispensable platform to study pulp biology, disease mechanisms, and novel therapeutic strategies.

## 2. Architectures and Strategies in 3D Dental Pulp Modeling: Biomaterials, Cells, and Advanced Culture Systems

Three-dimensional dental pulp models have become essential tools for exploring pulp biology and advancing regenerative endodontic therapies. These models vary widely in their composition, design, and dynamic properties, each bringing distinct insights and experimental possibilities.

### 2.1. Composition

The choice of biomaterial is central to any 3D pulp model, as it determines the microenvironmental cues that regulate cell behavior, differentiation, and matrix production. Natural polymers such as collagen, fibrin, gelatin, hyaluronic acid, alginate, and chitosan closely mimic the extracellular matrix and provide excellent biocompatibility and biodegradability. Their main limitation lies in their poor mechanical strength and variability. In contrast, synthetic polymers such as polyethylene glycol (PEG), polycaprolactone (PCL), and poly (lactic-co-glycolic acid) (PLGA) allow precise control of stiffness, porosity, and degradation rate but lack native biological signals. Hybrid systems that combine synthetic matrices with bioactive peptides or growth factors have been developed to merge the advantages of both classes of materials [10,11].

Matrigel remains one of the most widely used hydrogels for in vitro pulp studies because of its capacity to support complex 3D tissue formation, although its undefined composition and xenogeneic origin limit its translational use. Decellularized pulp-derived extracellular matrix hydrogels from porcine or human sources represent a promising alternative, as they retain many biochemical components of the native tissue. At the same time, 3D bioprinting technologies have opened the possibility of constructing scaffolds that reproduce root canal geometries or fine nanofibrillar architectures. Gelatin methacrylate (GelMA) and alginate are among the most commonly used bioinks for such applications, providing tunable mechanical and biological properties suitable for dental pulp regeneration [10,12].

The cellular component of 3D pulp models is equally crucial. Dental pulp stem cells (DPSCs) are the most commonly used population because of their accessibility, rapid proliferation, and ability to differentiate into odontoblasts, osteoblasts, and neural-like cells. Other mesenchymal stem cell (MSC) populations, such as stem cells from the apical papilla (SCAPs), periodontal ligament stem cells (PDLSCs), stem cells from exfoliated deciduous teeth (SHEDs), and dental follicle stem cells, are also used to model different developmental or regenerative contexts. The inclusion of endothelial cells, typically human umbilical vein endothelial cells (HUVECs), enhances angiogenic potential and promotes the formation of prevascular structures that improve tissue viability. Increasingly, multicellular co-cultures combining DPSCs with HUVECs, SCAPs, or macrophages are used to more closely reproduce the complex cellular heterogeneity and communication observed in vivo within the pulp [3,8] (Figure 1).

### 2.2. Scaffold-Based Constructs

Scaffold-based 3D models rely on porous matrices or fibrous networks composed of natural or synthetic biomaterials seeded with stem cells. Collagen sponges, silk fibroin scaffolds, polylactic matrices, and ceramics such as hydroxyapatite or β-tricalcium phosphate have all been used to generate pulp-like tissue both in vitro and in vivo. (Figure 1). These scaffolds allow precise control over geometry and pore size and are suitable for implantation studies, yet their fabrication can be technically demanding. The mechanical stiffness and degradation rate of the chosen material strongly influence cell differentiation and matrix deposition, and limited nutrient diffusion often restricts long-term culture viability in static systems [2,10,13,14]. Despite their utility, scaffold-based models often rely on lab-specific fabrication techniques and variable cell sources, which can limit reproducibility. Many studies are constrained to small sample sizes or short-term cultures, making it difficult to generalize findings or confirm functional outcomes across different laboratories.

### 2.3. Three-Dimensional Bioprinted Constructs

Three-dimensional bioprinting allows the layer-by-layer deposition of cells and biomaterials to produce organized pulp-like structures. Extrusion-based printing is most commonly employed in endodontic applications. For example, Duarte Campos and colleagues developed a collagen-based bioink containing DPSCs and HUVECs that successfully generated vascularized pulp tissue in vitro [15]. Recent advances in bioprinting have also enabled the fabrication of pulp-like constructs using bioinks enriched with DPSCs and factors such as vascular endothelial growth factor (VEGF). In one approach, a porous dental pulp guidance construct (DPGC) was printed using a GelMA-Dextran emulsion, which enhanced DPSC stemness. The microporous design supported cell adhesion, migration, and survival, while also encouraging neurogenesis and the formation of capillary-like networks. When these constructs were implanted subcutaneously in mice, they developed pulp-like tissue featuring odontoblast-like cells and newly formed vascular structures, closely mimicking native pulp architecture [16]. In addition to cell-laden bioprinted constructs, scaffold design plays a critical role in supporting hDPSC behavior. Comparative studies have shown that 3D-printed alginate/gelatin (Alg-Gel) scaffolds enhance cell adhesion and proliferation more effectively than conventionally cast counterparts. Moreover, aqueous extracts from these 3D-printed scaffolds further promote osteogenic and odontogenic differentiation, as indicated by increased formation of bone-like nodules, elevated alkaline phosphatase activity, and upregulation of mineralization-related genes. Elemental analysis revealed higher calcium and phosphorus content in the printed scaffolds, suggesting that the microenvironment they create is more conducive to hDPSC maturation and functional tissue formation [17]. Moreover, a study has shown that decellularized ECM derived from DPSCs can enhance 3D bioprinted pulp constructs by supporting cell adhesion, proliferation, and mineralization, even in non-mineralizing cells [18]. Key components, including collagens, and annexins, provide a bioactive microenvironment that can be integrated into bioprinted scaffolds to promote vascularized and functionally mature pulp-like tissues, potentially improving construct performance and accelerating pulp regeneration [19].

Bioprinting enables unparalleled spatial control, permitting the creation of complex geometries, multicellular gradients, and predesigned microchannels that favor vascular ingrowth. However, the technique requires specialized equipment, and the printing process can expose cells to mechanical shear stress that may affect their survival. The number of bioinks that combine printability with biological functionality remains limited, which constrains broader application [9,20].

Recent advances have begun to equip 3D bioprinted and 3D-printed materials with antimicrobial functionality, which is particularly relevant in regenerative endodontics where infection control is critical. Natural antimicrobial additives, such as plant and propolis extracts, can be incorporated into dental materials to inhibit common oral pathogens including *Streptococcus mutans*, *Enterococcus faecalis*, and *Candida albicans*, while maintaining low toxicity and biocompatibility [21]. In addition, 3D printing resins modified with fluoride-releasing complexes have been shown to continuously release fluoride ions over several weeks, demonstrating antibacterial efficacy against *S. mutans* without affecting cytocompatibility or mechanical performance [22]. These findings suggest that bioinks or printable scaffolds can be designed to support stem cell survival and vascular/nerve ingrowth while actively suppressing microbial colonization. Incorporating antimicrobial 3D-printed constructs into pulp-tissue engineering could enhance efficiency by reducing endodontic infection and improve functionality by setting up a healthy microenvironment, representing a promising direction for next-generation regenerative endodontic biomaterials [23,24,25,26,27,28,29,30,31]. In conclusion, although bioprinting enables precise spatial control, the number of bioinks with validated biological functionality is limited, and printing parameters vary widely among studies. Consequently, reproducibility of vascularized or innervated pulp constructs remains a concern, highlighting the need for standardized printing protocols and robust cross-laboratory validation.

### 2.4. Injectable Hydrogels

Injectable hydrogels are hydrophilic biomaterials that form soft, hydrated matrices closely resembling the mechanical environment of native dental pulp. They can be directly delivered into root canals and gel in situ, allowing minimally invasive filling of the pulp space while adapting to its complex three-dimensional geometry. Cells can be encapsulated within or seeded onto these hydrogels, providing a supportive microenvironment that promotes migration, proliferation, and differentiation.

Natural hydrogels such as collagen, gelatin, fibrin, and hyaluronic acid are widely used because of their inherent biocompatibility, biodegradability, and ability to support cell adhesion and signaling. Synthetic polymers, including polyethylene glycol (PEG), poly (lactic-co-glycolic acid) (PLGA), and polycaprolactone (PCL), allow precise control over mechanical stiffness, porosity, and degradation rate, enabling fine-tuning of the 3D microenvironment. Hydrogels can also be biofunctionalized with RGD adhesion motifs or loaded with growth factors such as TGF-β, VEGF, or BMPs to enhance odontogenic differentiation and angiogenesis [10,12].

Despite their advantages, injectable hydrogels generally exhibit low mechanical stability and rapid degradation, which can limit long-term culture, structural integrity, and support for large-volume constructs. Natural hydrogels may show batch-to-batch variability and limited inherent mechanical strength, while synthetic hydrogels often lack bioactivity unless functionalized. In addition, precise control of gelation kinetics and delivery into narrow confined root canal spaces can be technically so challenging, that crosslinking or reinforcement strategies may be required to ensure stability and durability in vivo [32] (see Table 1). To address these limitations, chemical crosslinking, composite formulations, or reinforcement with bioactive fillers are often employed. Athirasala and collaborators demonstrated that a dentin-derived alginate hydrogel could effectively guide stem cells from the apical papilla (SCAPs) toward dentinogenesis, highlighting the potential of biomimetic hydrogels for regenerative endodontic applications [33].

In summary, while injectable hydrogels provide a versatile and minimally invasive scaffold for 3D dental pulp models, supporting cell viability, differentiation, and paracrine signaling, variability in polymer batches, gelation kinetics, and bioactive factor incorporation contribute to inconsistent results. Many reports are based on single-lab experiments, underscoring the importance of systematic replication to ensure reproducible odontogenic and angiogenic outcomes. Nevertheless, when carefully designed and optimized, they offer tunable physical and biochemical properties that can be adapted for both preclinical modeling and regenerative therapies, while addressing inherent limitations of mechanical stability, bioactivity, and long-term structural integrity.

### 2.5. Spheroid Models

Spheroid models are simple, scaffold-free 3D aggregates formed under low-adhesion conditions, such as hanging-drop, ultra-low attachment plates, micropatterned wells, or magnetic levitation. These microtissues allow close cell–cell contact and recapitulate natural signaling networks, leading to enhanced extracellular matrix production and differentiation compared with 2D monolayer cultures. Spheroids can be monocellular or multicellular, and in dental pulp research, they serve dual purposes: as versatile in vitro models for studying pulp biology, stem-cell behavior, and biomaterial interactions and as mini-pulp constructs for potential implantation into root canals.

Cells within spheroid cores often experience hypoxia or necrosis, while peripheral cells remain proliferative, reproducing physiological oxygen and nutrient gradients that are absent in 2D cultures. This gradient effect mirrors conditions in native pulp tissue, adding physiological relevance to the model. Dissanayaka and colleagues generated spheroids from DPSCs and HUVECs that produced pulp-like extracellular matrix and exhibited early signs of vascular organization [34]. Compared with monolayer cultures, DPSC spheroids show significant differences in viability, phenotype, and differentiation potential, highlighting their value for studying paracrine signaling, early tissue assembly, and angiogenic interactions [35,36].

Despite their advantages, spheroids are limited by their lack of predefined architecture and restricted nutrient diffusion, which can result in necrosis in the central regions. Nevertheless, they remain a practical and physiologically relevant platform for investigating dental pulp regeneration and cellular responses in a controlled 3D environment [36].

### 2.6. Dentin-Pulp Organoids

Organoid technology represents an advanced evolution of 3D culture, in which stem or progenitor cells self-organize into microtissues that recapitulate both the structure and function of native pulp. Most dental pulp organoids are grown in hydrogels such as Matrigel or collagen, which provide a supportive basement membrane-like environment that mimics the extracellular matrix. Unlike simple spheroids, organoids exhibit greater tissue complexity, combining mesenchymal lineages such as DPSCs and SCAPs, and occasionally epithelial cells, to form structures resembling pulp and dentin.

DPSC-derived organoids have been shown to develop distinct odontoblast-like and fibroblast-like layers, and in some cases primitive vascular networks, enabling them to respond to calcium silicate capping agents in a manner similar to natural pulp. (Jeong et al. engineered dentin–pulp organoids expressing odontoblastic markers and used them to test biocompatibility with silicate-based cements [37], while Xu et al. developed 3D DPSC–endothelial organoids with human pulp matrix to evaluate cytotoxicity and material responses [38]. These models allow stable long-term culture and capture the cellular heterogeneity of native pulp, providing a highly realistic platform for studying tissue development, disease progression, and therapeutic interventions [39,40].

Beyond conventional hydrogel-based organoids, recent studies have explored the integration of 3D printing technology for pulp tissue fabrication. Using bioprinting approaches, DPSCs can be precisely deposited within bioinks composed of collagen, gelatin, or other ECM-mimetic hydrogels, enabling the creation of pulp-like structures with controlled architecture, spatial cell distribution, and microchannel networks for improved vascularization [41,42]. These 3D-printed pulp constructs not only support odontogenic differentiation but also allow systematic evaluation of biomaterials, scaffold designs, and regenerative strategies under conditions closely mimicking the in vivo microenvironment. By combining organoid self-organization with the structural precision of bioprinting, such approaches offer a promising route toward reproducible and scalable pulp tissue engineering for translational applications [43,44]. Combining bioprinting of DPSCs or organoids with hydrogels allows not only precise spatial organization but also the tuning of mechanical and structural cues, enabling mechanobiological analyses and the study of cell–matrix interactions within defined 3D environments, which may further enhance the maturation and functionality of engineered pulp tissue [45,46].

Beyond pulp-specific organoids, “tooth germ organoids” integrating multiple cell types have partially replicated early tooth development, including epithelial invagination, pointing toward the future possibility of complete bioengineered tooth formation. Despite their promise, organoids remain limited by reliance on animal-derived matrices and the absence of fully functional vascular and neural networks, which continue to pose challenges for clinical translation.

### 2.7. Dynamic Culture Systems

Static 3D cultures often face limitations in nutrient and oxygen diffusion, leading to central hypoxia and cell death. Dynamic culture systems, including perfusion bioreactors, rotating wall vessels, and magnetic stirring platforms, overcome these issues by promoting medium flow and providing mechanical stimulation such as shear stress and pressure. Perfusion bioreactors, in particular, enhance odontogenic differentiation, extracellular matrix deposition, and mineralization by simulating physiological fluid dynamics. Although more complex and costly than static cultures, these systems significantly improve tissue homogeneity, maturation, and functional development of dental pulp constructs, offering a closer approximation to in vivo pulp physiology.

### 2.8. Microfluidic “Tooth-on-a-Chip” Models

Microfluidic and lab-on-a-chip technologies represent the most advanced iteration of 3D pulp modeling. In these systems, dental pulp cells and dentin slices are integrated into microchannels, allowing continuous perfusion and recreation of the dynamic microenvironment of the tooth. Microfluidic “pulp-on-a-chip” platforms, including the model developed by França and colleagues, enable real-time observation of pulp cell responses to endodontic materials and controlled delivery of bioactive molecules. The architecture of microchannels separated by dentin fragments allows simulation of root canal perfusion and material diffusion, combining precise physical control with online analysis. These platforms provide regulation of fluid flow, nutrient gradients, and mechanical stimuli, enabling studies of the dentin-pulp interface under near-physiological conditions [47].

Together, dynamic bioreactors and tooth-on-a-chip systems provide complementary approaches to study pulp physiology, pathology, and regenerative strategies. While bioreactors improve tissue maturation and homogeneity, microfluidic platforms allow high-resolution mechanistic studies, material testing, and modeling of the root canal microenvironment. The integration of these technologies exemplifies the convergence of stem cell biology, biomaterials science, and microengineering, advancing 3D pulp models toward clinically relevant applications in regenerative endodontics (Figure 2).

## 3. Harnessing 3D Pulp Models for Therapeutic Innovation

### 3.1. Modeling Physiologic Pulp Responses

Three-dimensional dental pulp models provide a versatile platform to investigate physiological processes within the dentin–pulp complex, offering insights that are difficult to obtain from traditional 2D cultures. In studies of physiologic pulp, 3D constructs enable detailed evaluation of interactions between dental cells and biomaterials. They allow researchers to assess biocompatibility by monitoring cell viability, proliferation, and apoptosis in response to new restorative materials, while also probing odontogenic differentiation through expression of dentin-specific markers such as DSPP, DMP-1, ALP, and Runx2. These models further support angiogenesis and neurogenesis studies by integrating endothelial or neural cells, examining tube formation, VEGF expression, and responses to neurotrophic factors. Inflammatory responses can also be characterized, for instance by measuring cytokine release, such as IL-6 or TNF-α, following exposure to restorative agents. Notably, Xu et al. developed a DPSC–endothelial organoid incorporating human pulp ECM to evaluate pulp responses to lipopolysaccharide or pulp-capping cements: LPS induced inflammatory cytokines (IL-6, IL-1β) and cell apoptosis as in pulpitis [38], while a bioceramic (iRoot BP) led to calcium deposition. Such 3D models better mimic the pulp’s drug response than 2D cells. Similarly, “tooth-on-a-chip” microfluidic devices can simulate the dentin barrier and pulp perfusion for toxicology studies [47].

### 3.2. Simulating Pathologic Pulp Conditions

Beyond physiological studies, 3D pulp models are increasingly applied to simulate disease processes in vitro. Pathological conditions such as pulpitis, calcific metamorphosis, and other inflammatory states can be reproduced by exposing organoids, spheroids, or co-cultures to bacterial lipopolysaccharides (LPSs) or oxidative stress, leading to cell death and the upregulation of inflammatory markers [4]. These models can also incorporate immune cells, such as macrophages, to investigate infection dynamics, cytokine signaling, and inflammatory cascades [9,38].

Advanced organoid and organ-on-chip platforms allow precise control of environmental, biochemical, and mechanical parameters, enabling the study of pulp inflammation, neovascularization, neurogenesis, and tissue repair under physiologically relevant conditions. “Tooth-on-a-chip” systems further simulate pulp–dentin–vascular interactions with dynamic fluid flow and mechanical stimulation, providing a robust tool for testing targeted therapies, biomaterials, and pharmacological interventions in biologically relevant microenvironments. While still at an early stage, these disease modeling approaches are rapidly expanding, offering unprecedented opportunities to dissect pathological mechanisms and evaluate innovative therapeutic strategies [3,47,48,49,50,51,52,53,54,55,56,57,58,59,60,61,62,63,64,65,66,67,68,69,70,71] (Figure 3, Table 2).

### 3.3. Advancing Regenerative Endodontics

Regenerative endodontics represents one of the most transformative applications of 3D pulp models, providing a versatile platform to study and develop strategies for functional pulp regeneration in treated teeth. These systems enable the in vitro investigation of DPSC differentiation, odontoblast activity, and dentin formation, as well as optimization of chemotactic gels or growth-factor–loaded scaffolds for clinical translation. Approaches include pulp capping to preserve vitality and cell-homing therapies for necrotic canals [10]. Semi-orthotopic tooth slice or root canal models seeded with DPSCs and scaffold have demonstrated formation of new dentin and vascularized pulp tissue. For instance, Suzuki et al. employed collagen gels containing SDF-1 and bFGF to recruit DPSCs and generate pulp tissue in a dog model [48].

Clinical pilot studies have begun translating these findings, with autologous DPSC grafts achieving pulp regeneration in human teeth, as shown by Xuan et al. [49]. These models serve both as research tools for elucidating pulp biology and as preclinical platforms guiding the development of scaffold-based and cell-based regenerative therapies, bridging the gap between laboratory experimentation and clinical application [40].

### 3.4. Tissue Engineering and Biofabrication of Pulp

Tissue engineering and biofabrication represent the ultimate goal of creating transplantable pulp tissue. Injectable cell-hydrogel constructs have been applied to animal root canals [9,50]. Bioprinted scaffolds have also been used: Duarte Campos et al. implanted a collagen-DPSC/HUVEC bioink in swine teeth and achieved complete pulp revascularization. These models advance clinical translational research by demonstrating functional tissue formation [15]. These studies show that 3D dental pulp models are powerful tools for dissecting fundamental biological processes, evaluating biomaterials, and designing next-generation regenerative therapies. They bridge the gap between in vitro experimentation and clinical application, opening avenues for safer and more effective endodontic therapies [73,74,75,76,77,78,79,80,81,82,83,84,85,86,87,88,89,90]. (Figure 3, Table 3).

## 4. Future Perspectives and Unmet Challenges in 3D Pulp Modeling

### 4.1. Harmonizing Stem Cell Sources and Culture Conditions

A major limitation in 3D pulp modeling is the heterogeneity of dental stem cell sources. Donor variability, including age, tooth type, and systemic health, as well as phenotypic drift during in vitro expansion, can significantly affect proliferation, differentiation potential, and reproducibility. Differences in isolation techniques and culture media further amplify variability between studies. Standardized protocols are urgently needed, alongside defined, chemically reproducible matrices such as synthetic hydrogels instead of Matrigel, to ensure more consistent results across laboratories and improve comparability of experimental outcomes [9,50].

### 4.2. Replicating the Native Pulp Microenvironment

The dental pulp is exposed to a complex microenvironment, including mechanical loading, confinement within rigid dentin walls, vascular perfusion, and microbial interactions. Most current 3D models, however, fail to fully reproduce these conditions. Advanced organ-on-chip platforms and microfluidic systems are emerging as promising tools, as they allow dynamic perfusion, controlled fluid shear, and integration of biochemical cues. Such systems provide real-time monitoring of cell behavior and interactions, offering more physiologically relevant platforms to study both healthy and diseased pulp tissue.

### 4.3. Engineering Vascularization and Functional Innervation

Maintaining cell viability in thick 3D constructs remains a significant challenge due to limited nutrient diffusion and hypoxia in central regions [91]. Co-culture approaches with endothelial cells to pre-form microvessels, combined with integration of neural cells, are essential to promote vascularized and innervated pulp-like tissue [36]. Establishing functional innervation within 3D dental pulp cultures is pivotal for recapitulating the physiological microenvironment of dentinogenesis. Beyond its sensory function, pulpal innervation exerts crucial trophic, regulatory, and inductive effects that govern dentin formation, homeostasis, and repair. Perfusion using bioreactors or microfluidic chips further supports tissue maturation, while hydrogel-based delivery scaffolds enhance cell localization, viability, and regenerative potential. Lessons from other regenerative fields demonstrate that embedding stem cells in bioactive hydrogels or growth factor–loaded matrices can significantly improve tissue survival and functionality [13,36,92,93].

### 4.4. Addressing Reproducibility and Standardization

Beyond stem cell variability, differences in scaffold composition, organoid size, and culture conditions contribute to poor reproducibility. Many published studies are limited by small sample sizes, lab-specific protocols, and reliance on animal-derived matrices, reducing confidence in their generalizability. Batch-to-batch variability in materials such as Matrigel, as well as inconsistent differentiation protocols, limits the ability to compare results across studies. Development of synthetic, well-defined matrices, alongside standardized evaluation criteria for cell viability, odontogenic differentiation, angiogenic potential, and immune competence, is essential for generating reliable, reproducible models suitable for translational research [50].

### 4.5. Integrating Multi-Cellular Complexity and Immune Interactions

Pulp tissue is not solely composed of dental stem cells and fibroblasts; it contains, endothelial cells, neural cells, and immune populations such as macrophages. Incorporating these components into 3D models is necessary to study multi-cellular interactions, inflammation, and pain signaling. Such integration enables the modeling of both physiological and pathological processes, including pulpitis, infection, and regenerative responses, providing a completer and more predictive platform for preclinical studies.

### 4.6. Bridging Translation to the Clinic

Translating 3D pulp constructs into clinically relevant regenerative therapies requires addressing multiple interrelated challenges while implementing practical strategies to overcome them. Whatever their composition or fabrication strategy, the engineered 3D pulp should be easily and reproducibly inserted into the endodontic canals of all teeth, including those with pronounced curvatures, and should ensure effective pulp regeneration once implanted (Figure 2). Thus, scaling constructs to dimensions relevant for implantation in narrow endodontic spaces demands GMP-compliant biomaterials, standardized potency assays, and robust evaluations of both mechanical and biological properties, encompassing dental stem cell “fitness”, including their angiogenic, odontogenic, and immunomodulatory capacities. To address donor variability, rigorous cell characterization, banking of well-defined cell populations, and the use of allogeneic or iPSC-derived standardized cells can also improve reproducibility and clinical consistency [94,95,96].

Cell-free approaches, such as extracellular vesicles or secretome-based therapies, and hybrid Advanced Therapy Medicinal Products (ATMPs) integrating implantable medical devices with advanced cell- or biologic-based constructs, offer complementary strategies to enhance immune compatibility, paracrine signaling, and survival in the root canal microenvironment. From a regulatory perspective, cell-free biologics are generally considered simpler to translate than living ATMPs, as they do not contain viable cells, carry a lower risk of tumorigenicity, and may be classified outside the strictest ATMP regulatory pathways in certain jurisdictions [97,98] (see Table 4). Nevertheless, appropriate quality control, potency assays, and characterization of bioactive components remain essential [98,99].

Additionally, advanced delivery platforms, including injectable hydrogels, bioprinted scaffolds, and microfluidic devices, are critical to enhance localization, vascularization, and regenerative durability [9].

Combined ATMP provide enhanced functional outcomes by uniting scaffold support, controlled delivery of bioactive factors, and/or cellular components, but require additional regulatory considerations including device biocompatibility, mechanical performance, sterilization, and interface with the biologic component.

Manufacturing constraints can be mitigated through modular scaffold designs, automated bioprinting, and rigorous quality control, ensuring reproducibility and scalability. Optimization of material composition, mechanical properties, and controlled release of bioactive cues enhances integration and long-term stability. Microfluidic platforms and organ-on-chip systems can serve as predictive preclinical models to test these constructs under physiologically relevant conditions, informing design before in vivo application [100].

Regulatory translation requires early engagement with regulatory agencies, compliance with tissue-engineering and cell therapy guidelines, and GLP-compliant preclinical studies to assess safety, immunogenicity, biodistribution, and long-term efficacy. Establishing standardized potency assays, functional benchmarks, and preclinical endpoints aligned with regulatory expectations accelerates approval pathways.

Ultimately, bridging the gap from bench to clinic demands an integrated strategy combining optimized biomaterials, reproducible cell sources, advanced delivery systems, predictive preclinical models, and regulatory planning. Interdisciplinary collaboration among bioengineers, stem cell biologists, clinicians, and regulatory experts is essential to transform 3D pulp constructs into safe, effective, and clinically translatable regenerative endodontic therapies. By simultaneously addressing challenges and implementing targeted solutions, these strategies provide a clear roadmap for moving next-generation pulp regeneration from experimental models to patient care (Table 4).

### 4.7. Toward Next-Generation 3D Pulp Models

Addressing these gaps requires an integrated strategy combining standardized stem cell sources, faithful replication of the pulp microenvironment, robust vascularization and innervation, and advanced multi-cellular models. The convergence of organ-on-chip technologies, bioactive scaffolds, and innovative delivery systems will be pivotal in transforming 3D pulp models from experimental research tools into reliable, translational platforms for regenerative endodontics. Such advances will not only improve mechanistic understanding of pulp biology but also accelerate the development of clinically viable, evidence-based therapies for dental pulp.

## 5. Advancing 3D Dental Pulp Models

Three-dimensional (3D) in vitro models of dental pulp have rapidly evolved from simple cell aggregates to complex organoid and microfluidic systems that recapitulate native tissue architecture, cellular diversity, and function. Moving beyond two-dimensional (2D) monolayers, 3D environments restore physiological phenotypes of dental pulp stem cells (DPSCs) and enable modeling of both healthy and pathological pulp states.

Early spheroid studies demonstrated that DPSCs in 3D matrices such as Matrigel or collagen form organized microtissues with enhanced odontogenic marker expression (ALP, DSPP, DMP-1) and mineralization compared with 2D culture. These spheroids capture critical cell–cell and cell–matrix signaling, providing a more physiologically relevant microenvironment [59,66].

Advances in biomaterials have expanded modeling possibilities. Natural hydrogels like Matrigel and collagen support organoid formation [37,38], while synthetic scaffolds such as self-assembling peptides (Puramatrix™) or hybrid GelMA-alginate composites enable mechanical tunability and 3D bioprinting [73,90]. Decellularized pulp ECM preserves structural and biochemical cues that enhance stem-cell adhesion and odontoblastic differentiation [100]. Scaffold-free approaches relying on DPSC aggregation further allow intrinsic tissue organization without exogenous materials [101].

Reproducing pulp complexity has been achieved through multicellular co-cultures. DPSCs with endothelial cells (HUVECs) promote angiogenesis and vessel-like formation [67,72], while more sophisticated constructs integrate immune cells to recreate inflammatory or reparative microenvironments [40]. Prevascularized organoids under hypoxic and paracrine conditions have been analyzed via single-cell RNA sequencing, confirming the essential role of vascular and immune crosstalk for realistic pulp regeneration [36].

Microfluidic platforms, including dentin-on-a-chip systems, reproduce the dentin–pulp interface with microchannels that mimic dentinal tubules and support odontoblast processes [51,52]. These devices replicate fluid flow and nutrient gradients, providing robust platforms for biomaterial testing, drug screening, and mechanobiology studies.

Clinically oriented models aim to regenerate functional pulp. Scaffold-free DPSC sheets and prevascularized organoids implanted into animal root canals generate pulp-like tissue with blood vessels and odontoblast layers [38]. Decellularized pulp scaffolds seeded with DPSCs or SCAPs reconstruct dentin–pulp complexes in vivo, demonstrating translational potential [83,100,101].

Mechanistic insights reveal consistent roles for HIF-1α and VEGF in angiogenesis, and BMP, Wnt, and TGF-β signaling in odontoblastic differentiation. Emerging strategies using epigenetic modulators or nanoplatforms such as BMP-releasing microparticles or DNMT-inhibitor-loaded scaffolds further refine lineage specification and tissue development [60,70,102]. Integration of omics approaches now enables dissection of microenvironmental heterogeneity and differentiation pathways within organoids.

Future directions emphasize convergence of tissue engineering, biomaterials, and organ-on-chip technologies. ey objectives include 3D bioprinting of composite pulp–periodontal constructs with spatially controlled biofactor gradients, incorporation of neuronal components to reproduce neurovascular regulation and pain perception, and patient-specific iPSC-derived organoids for personalized drug and biomaterial testing. Standardization, reproducibility, and cost-effective matrices remain critical for clinical translation. Three-Dimensional dental pulp models now span spheroids, vascularized organoids, and microfluidic chips, bridging in vitro experimentation and in vivo functionality. These bioengineered systems are poised to transform regenerative endodontics by providing predictive, mechanistic, and translationally relevant platforms for next-generation therapies.

## 6. Conclusions and Future Perspectives: Towards Predictive and Translational 3D Dental Pulp Models

Recent years have witnessed significant advances in 3D dental pulp culture, highlighting the emergence of organoid and microengineered platforms as essential tools for regenerative endodontic research. Dental organoid models, including pulp, dentin, and tooth germ organoids, have been successfully developed to recapitulate native tissue architecture and cellular heterogeneity. Notably, Jeong and Xu established pulp organoids expressing odontoblastic markers, which have been used to evaluate biomaterial biocompatibility and study differentiation dynamics [37,38,64].

In parallel, 3D bioprinting technologies have matured, enabling the precise fabrication of scaffolds that replicate complex root canal geometries and microenvironments. The integration of multicellular co-cultures, combining pulp cells with macrophages or endothelial networks, alongside microfluidic tooth-on-chip systems, demonstrates the field’s drive to reproduce the in vivo complexity of the dental pulp niche [103]. These platforms allow controlled perfusion, dynamic signaling, and the real-time monitoring of cellular responses, enhancing the predictive accuracy of in vitro assays for biomaterial testing, disease modeling, and mechanistic studies.

Different classes of materials currently employed for pulp restoration exhibit both advantages and limitations that influence their regenerative performance. Natural polymers such as collagen or fibrin provide favorable biological compatibility and cell adhesion but have limited mechanical strength and stability. Synthetic polymers offer better structural properties and controlled degradation, yet may lack bioactivity or cellular integration. Combining these materials within composite or hybrid systems has shown promise in balancing biological and mechanical requirements, though reproducibility and standardization remain ongoing challenges [10].

The integration of multiple advanced 3D pulp culture systems promotes substantial optimization. Combining the bioprinting of dental pulp stem cells (DPSCs) or organoids with hydrogels proves to be highly advantageous [46]. This synergistic combination of 3D culture models mitigates the inherent limitations of each system while harnessing their respective advantages. Consequently, the advancement and refinement of future 3D pulp culture models are expected to rely on the strategic combination of complementary methodologies.

Current trends in the field emphasize increasing model complexity. Induced pluripotent stem cells (iPSCs) are being employed to generate multilayered organoids, while strategies for in vitro vascularization, neural integration, and dynamic mechanical or biochemical signaling are implemented to more faithfully mimic physiological and pathological conditions. Future studies should focus on optimizing material–cell interactions within these advanced 3D systems, improving reproducibility, and developing standardized models that more accurately predict clinical outcomes. Particular attention should be given to disease- and age-specific pulp models, integration with immune and vascular components, and the identification of novel biomarkers that can guide material selection and therapeutic design. In the future, building on these substantial advances, in vitro pulp models could be further refined to reproduce more specific and complex conditions. These may include models interacting with various pathogenic biofilms, models of different endo-periodontal lesions, senescent pulp models with reduced regenerative potential, and models for studying endodontic pain control. Furthermore, such models could support not only therapeutic innovations but also diagnostic developments, by reproducing the various stages of pulpal inflammation to identify new biomarkers and establish more precise diagnostic criteria for assessing pulp status and its regenerative potential.

These advancements are transforming 3D dental pulp models into robust and predictive platforms that closely reflect native pulp biology, enabling reliable evaluation of restorative materials and regenerative therapies while guiding the design of next-generation scaffolds and translational treatment strategies.

By bridging fundamental pulp biology with clinical applications, these next-generation 3D models hold the potential to accelerate the discovery and validation of novel dental treatments, optimize biomaterial design, and ultimately support the realization of functional, patient-specific regenerative endodontic therapies.

## Figures and Tables

**Figure 1 ijms-26-10960-f001:**
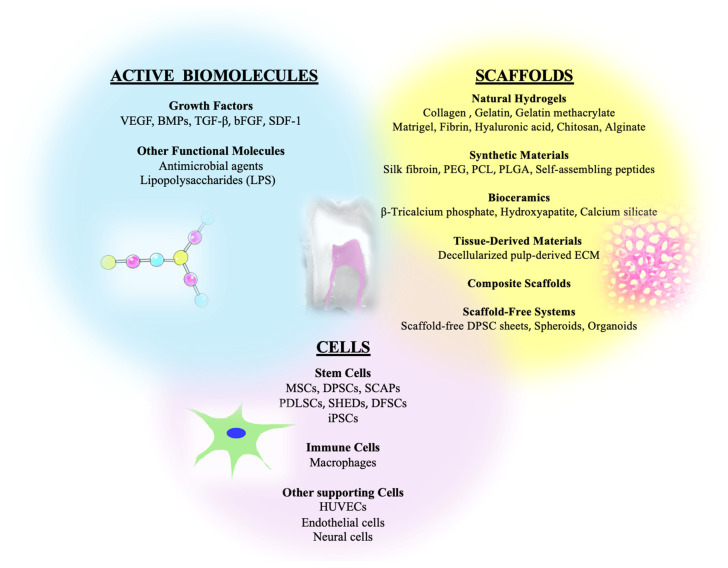
Key components for engineering three-dimensional dental pulp models. Active biomolecules (blue), scaffolds (yellow), and cells (purple) represent the essential building blocks for constructing 3D dental pulp culture systems and regenerative constructs.

**Figure 2 ijms-26-10960-f002:**
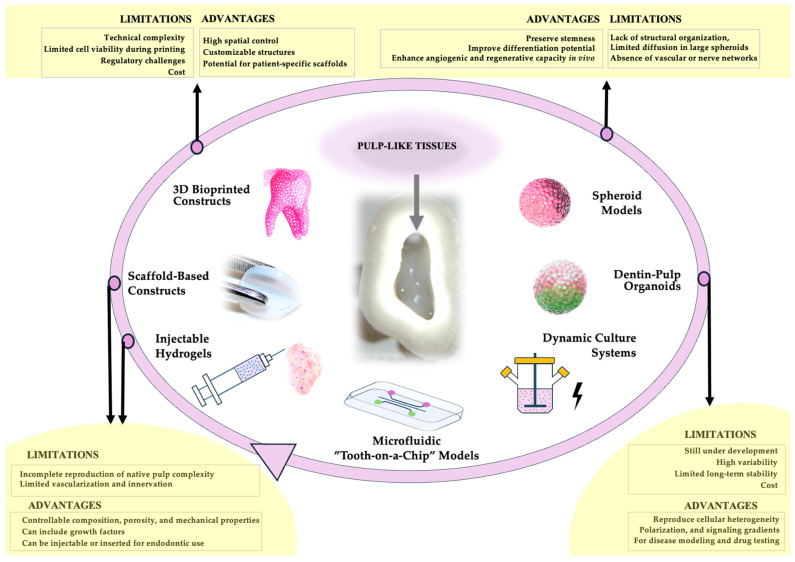
Progression of strategies for 3D dental pulp culture, from early hydrogel injection techniques to the development of sophisticated ‘tooth-on-a-chip’ platforms with dentin-like tissues (green). Synthesis of advantages and limitations for the main 3D dental pulp culture systems (yellow). These advanced culture systems allow the engineering of pulp-like tissues (pink) for insertion into confined endodontic spaces to promote regeneration.

**Figure 3 ijms-26-10960-f003:**
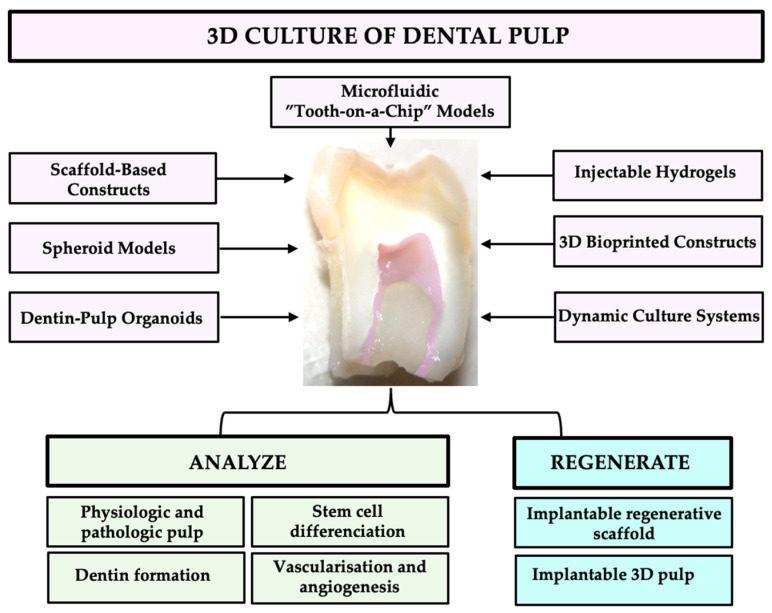
Different 3D culture of dental pulp (pink) and their applications for analyzing (green) and for regenerating (blue) 3D pulp tissues.

**Table 1 ijms-26-10960-t001:** Key features, advantages, and limitations of injectable hydrogels for 3D dental pulp models and regenerative endodontics.

Feature	Advantages	Disadvantages
Injectability & Handling	Minimally invasive delivery Adapts to complex root canal geometry	Gelation and placement in confined spaces can be challenging
Cell support & Biocompatibility	Supports cell adhesion Proliferation and differentiation Biodegradable	Natural hydrogels may show batch variability Synthetic hydrogels require functionalization
Mechanical properties	Tunable stiffness Can be reinforced or crosslinked	Low intrinsic strength Rapid degradation limits long-term structural integrity
Bioactivity & Functionalization	Can incorporate growth factors (TGF-β, VEGF, BMPs) to enhance regeneration	Requires careful design to maintain stability and bioactivity
Adaptability & Versatility	Suitable for preclinical models and regenerative applications	Large-volume constructs may be difficult to maintain without reinforcement

**Table 2 ijms-26-10960-t002:** Three-Dimensional culture of dental pulp for analyzing physiologic and pathologic pulp, dentin formation, stem cell differentiation and vascularization.

3D CULTURE OF DENTAL PULP: TO ANALYZE		
Reference	Description	Interest
Physiologic and Pathologic Pulp		
[37]	Dentin–pulp-like organoids with stem/odontoblast features hDPSCs embedded in Matrigel	Mechanistic studies and testing pulp-capping biomaterials (Biodentine)
[6]	Tissue engineering-based Dentin/Pulp tissue analogue	Advanced biocompatibility evaluation tool of dental restorative materials
[38]	Pulp organoid with vessel-like structures, odontogenic & endothelial marker expression hDPCs + endothelial cells ± hDP-ECM	Biocompatibility/toxicity screening and vascularization modeling in pulp regeneration
[51]	Microfluidic chip mimicking dentin tubules Odontoblasts extend processes into channels	Odontoblast physiology, dentin–pulp interactions, and material screening
[52]	Microfluidic dentin-on-chip	High-throughput testing of dental materials and pulp–dentin interactions
[53]	Human dental pulp cells (hDPCs) spheroids showed ECM gene upregulation and collagen-rich matrix formation	ECM organization and deposition in pulp
[54]	DPCs in 3D culture model	Combined catalytic strategies applied to in-office tooth bleaching
[55]	Chitosan-based scaffold (animal vs fungal sources) co-polymerized with gelatin and crosslinked with GPTMS or genipin	Assessment of chitosan sources and biocompatibility with human DPSCs
[56]	Scaffold-free, collagen self-assembly with (SCAPs) and macrophages Forms cap-shaped apical papilla-like organoid	Periapical biology, disease environment, and therapeutic testing
[57]	Tooth root organoid Co-culture DPSCs + (PDLSCs)	Tooth root and pulp regeneration
[22]	3D dental pulp cell microtissues and *S. mutans*	Model of pathologic pulp
[58]	Cytotoxicity of filling materials on 3D pulp model	Model of pulp floor perforation
**Stem Cell Differentiation**		
[59]	DPSCs in Matrigel and other hydrogels Formation of mineralized nodules osteo/odontogenic markers expressed	DPSC differentiation and mineralization in 3D matrices
[60]	DPSCs in microsphere-forming plates Multilineage differentiation capacity enhanced versus 2D culture	Accessible 3D platform to evaluate DPSC regenerative potential
[40]	Influence of the microenvironnement (ECM, growth factors) on odontogenic differenciation of MSC	Effet of ECM on differenciation MSC and formation of dentin-pulp
[61]	Organoïdes of dental germ made by microparticules and hydrogel, guiding cellular agregation	Dental developpement
[62]	DPSC-MSC spheroids: higher ALP, DSPP, osteocalcin expression versus 2D culture Rapid mineralized nodule formation	Differentiation potential of DPSC-MSC
[63]	Organoids Dental pulp stem cells (DPSCs)	Potential regenerative of DPSCs
[64]	3D spheroid culture of dental pulp-derived stromal cells	Regenerative properties for therapeutic applications
[65]	Photobiomodulation therapy for DPSCs differenciation	Multilineage differentiation of DPSCs
**Dentin Formation**		
[66]	Mouse dental papilla cell spheroids; expression of ALP, DSPP, DMP-1 Analyse of mineralized nodules	Model odontoblast differentiation and dentinogenesis in 3D spheroids.
[67]	DPSCs + HUVECs co-culture in matrigel; enhanced odontogenic differentiation and vascular-like structures	Show synergistic effects of DPSCs + ECs in angiogenesis and odontogenesis.
[68]	3D explant culture of human dental pulp tissue in matrigel	Physiology of Pre-odontoblast
[69]	DPSC spheroids loaded with ZIF-8 nanoplatform releasing DNA methyltransferases inhibitors	Enhance odontogenesis through sustained epigenetic modulation
[70]	3D Spheroid Formation Using BMP-Loaded Microparticle Human–Differenciation DPSCs	Odontoblastic Differentiation
[71]	Combination of 3D Printing and ALD with DPSC	Dentin Fabrication
**Angiogenesis and Vascularisation**		
[34]	DPSC -HUVEC co-culture spheroids in Matrigel with growth factor supplementation	Study prevascularization and vascular network formation for pulp regeneration
[36]	Spheroid organoids of hDPSCs + ECs under hypoxia; characterized by scRNA-seq	Model angiogenesis and odontoblastic differentiation pathways
[72]	SCAPs + ECs under hypoxia Formed stable vascular-like networks EphrinB2 signaling involved	Model angiogenesis & vascular stabilization for pulp revascularization strategies.
[52]	Neovascularization by DPSC-ECs in a Tube Model	Model of Neovascularization

**Table 3 ijms-26-10960-t003:** Three-Dimensional culture of dental pulp for building adequate scaffolds and dental pulp tissues for endodontic regeneration.

3D CULTURE OF DENTAL PULP: TO REGENERATE		
Reference	Description	Interest
Implantable Adequate Scaffold		
[73]	Puramatrix™ self-assembling peptide hydrogel, DPSCs Viability and differentiation maintained	A synthetic peptide hydrogel as a 3D injectable scaffold for pulp regeneration
[15]	0.2% collagen type I + 0.5% agarose; inkjet bioprinted with human DPSCs and HUVECs	Formation of vascularized pulp-like networks ex vivo
[74]	Beta-TCP/PLGA (75:25) composite scaffold	Supporting osteo/odontogenic differentiation of DPSCs
[75]	Calcium silicate + PCL scaffold	Inducing odontogenic differentiation of DPSCs
[76]	GelMA conjugated with BMP-peptide, 3D bioprinted with DPSCs	Enhancing odontogenic differentiation and mineralization
[77]	GelMA with mineral trioxide aggregate (ProRoot MTA, Endosem Zr)	Promoting odontogenic differentiation of DPSCs
[78]	Injectable Double-Network Hydrogel	3D injectable scaffold for pulp regeneration
[79]	ECM from bone combined with β-TCP scaffold	Promoting osteo/odontogenic differentiation of DPSCs
[80]	Treated dentin matrix (TDM) + 30% PCL composite	Supporting odontogenic differentiation of dental follicle stem cells
[33,81]	Demineralized dentin matrix (DDM) mixed 1:1 with alginate and soluble dentin proteins	Inducing SCAPs differentiation into dentin-like structures
[82]	PLA/HA scaffold seeded with DPSCs	Promoting dentin-pulp-like tissue mineralization
[83]	Decellularized dental pulp extracellular matrix reseeded with SCAPs	Natural ECM scaffold for pulp tissue regeneration
[84]	Injectable decellularized dental pulp matrix-functionalized hydrogel microspheres	Scaffold for pulp regeneration
**Implantable Dental Pulp Tissue**		
[85]	Scaffold-free DPSC sheet/aggregate constructs, implanted into tooth root canals Pulp-like tissue with vasculature formed in vivo	Regenerate vascularized dental pulp tissue in vivo
[38]	Prevascularized pulp organoids formed from hDPSCs and HUVECs in Matrigel	Develop a physiologically relevant vascularized pulp organoid for regeneration
[86]	Microspheres of stem cells from human exfoliated deciduous teeth	Pulp regeneration capacity
[87]	Angiogenesis potential of self-assembled mesenchymal stem cell spheroids by size mediated physiological hypoxia	For vascularized pulp regeneration
[54]	Repopulation of a 3D simulated periapical lesion cavity with dental pulp stem cell spheroids with triggered osteoblastic differentiation	Regeneration of periapical lesion
[88]	Fabrication and characterization of 3D-printed polymeric-based scaffold coated with bioceramic, naringin and nHA with hDPSCs	Potential use in dental pulp regeneration
[89]	Vascularized DPSC constructs by inducing endothelial differentiation or co-culture strategies; constructs show perfusable/vessel-like structures in vitro	Provide prevascularized constructs to improve graft survival and accelerate in vivo angiogenesis after transplantation for pulp/bone applications
[90]	GelMA–alginate–bioactive glass microsphere bioink + stem cells for 3D bioprinting of pulp constructs	Scalable 3D bioprinting of pulp/periodontal regenerative constructs

**Table 4 ijms-26-10960-t004:** Comparison of key features, regulatory considerations, and functional attributes of cell-free biologics, implantable medical devices, ATMPs, and combined ATMPs for dental pulp regeneration.

Feature	Cell-Free Biologics	Implantable Medical Devices	ATMPs	Combined ATMPs
Composition	Extracellular vesicles Secretome	Scaffold materials 3D-printed constructs	Living cells (stem/progenitor)	ATMPs integrated with implantable medical devices
Regulatory complexity	Moderate Less stringent than ATMPs	High Must meet high standards (biocompatibility, sterilization, etc.)	High Full ATMP regulatory requirements	Very high Must meet both ATMP and medical device regulations
Safety considerations	Low tumorigenicity risk Lower immunogenicity	Device-related safety (mechanical stability, biocompatibility, degradation)	Potential tumorigenicity Immune rejection risk	Combination of both
Manufacturing	Scalable Standardized	Scalable fabrication Sterilization Quality control of scaffold/device	Complex Batch-to-batch variability	Complex Reproducibility of both biological and device components
Storage Logistics	Easier Stable formulations	Generally stable Storage depends on material type and sterilization requirements	Cryopreservation required Limited shelf-life	Depends on both ATMP and device May require specialized storage
Functional integration	Paracrine effects Immunomodulation	Structural support Guiding tissue regeneration Facilitating delivery for biologics/ cells	Cellular and paracrine activity for enhanced tissue repair Differentiation	Structural support Cellular and paracrine activity for enhanced tissue repair

## Data Availability

No new data were created or analyzed in this study. Data sharing is not applicable to this article.

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
