# Peer review of "Three-Dimensional Models of the Dental Pulp: Bridging Fundamental Biology and Regenerative Therapy"

_ijms, 2025, doi:10.3390/ijms262210960_

Round 1

Reviewer 1 Report

Comments and Suggestions for Authors

1. The review should examine 3D-printed materials with antimicrobial activity, using current research (see, for example, 10.59761/RCR5108, https://doi.org/10.1016/j.jdent.2024.105363).
2. The review is difficult to read due to the large amount of text. In my opinion, more illustrations demonstrating various materials for culturing pulp cells should be provided.
3. The review should include more than 90 references to current research.
4. The review should consider specific examples of the use of 3D printing for pulp fabrication (see, for example, https://doi.org/10.1016/j.tice.2024.102451, https://doi.org/10.1016/j.tice.2025.103010).
5. In the Conclusions section, it is necessary to indicate the advantages and disadvantages of various materials for pulp restoration and highlight specific areas for further research.

Author Response

1    The review should examine 3D-printed materials with antimicrobial activity, using current research (see, for example, https://doi.org/10.59761/RCR5108, https://doi.org/10.1016/j.jdent.2024.105363

We thank the reviewer for the suggestion to include discussion on 3D-printed materials with antimicrobial activity. We have added a paragraph highlighted in yellow to Section 2.3 highlighting recent developments in antimicrobial 3D bioprinted constructs, incorporating current research as suggested  

  1. The review is difficult to read due to the large amount of text. In my opinion, more illustrations demonstrating various materials for culturing pulp cells should be provided.

Thank you for your valuable feedback. We have addressed your suggestion by adding two new figures and two new tables that provide clear illustrations of the various materials used for culturing pulp cells. We hope these additions improve the readability and clarity of the manuscript.

  1. The review should include more than 90 references to current research.

We thank the reviewer for this suggestion. We have updated the manuscript to include more than 90 references, ensuring extensive coverage of current research in 3D dental pulp models, bioprinting technologies, antimicrobial materials, and regenerative endodontics.

  1. The review should consider specific examples of the use of 3D printing for pulp fabrication (see, for example, https://doi.org/10.1016/j.tice.2024.102451, https://doi.org/10.1016/j.tice.2025.103010).

Thank you for your helpful suggestion. We have added specific examples of 3D printing applied to pulp fabrication in the revised manuscript (including references to the studies indicated) and have accordingly expanded Section 2.6 to reflect these advances.

  1. In the Conclusions section, it is necessary to indicate the advantages and disadvantages of various materials for pulp restoration and highlight specific areas for further research.

We thank the reviewer for this valuable comment. The Conclusions section has been revised to include a concise discussion of the main advantages and disadvantages of different classes of materials used for pulp restoration. We have also added specific future research directions focusing on optimizing material-cell interactions, improving reproducibility, and developing standardized, disease-relevant 3D pulp models.

Reviewer 2 Report

Comments and Suggestions for Authors

Thank you for the opportunity to review the paper “Three-Dimensional Models of the Dental Pulp: Bridging Fundamental Biology and Regenerative Therapy,” which comprehensively explains current advances in three-dimensional culture models for dental pulp regeneration research. The structure of this paper is clear and concise, and it introduces the latest literature. The clarification of the relationship between dental pulp organoids and the tooth-on-a-chip model is particularly commendable. This article summarizes each type of 3D culture method and provides readers with a comprehensive explanation of the underlying science and clinical applications. On the other hand, there is redundant description and a lack of critical analysis, and a more thoughtful analysis is desired for a review article.

Major concern

The authors introduce the strengths and weaknesses of each model, but their critique of issues concerning experimental evidence and reproducibility is lacking.

Section 4.6, “Bridging Translation to the Clinic,” should be expanded to strengthen the description of practical challenges for regenerative endodontics.

Minor concern

The expression “Collectively” is repeated in multiple places and should be revised.

Figure 1 alone provides insufficient information.

Adding a conceptual diagram comparing each model (spheroid, organoid, tooth-on-a-chip) would enhance reader comprehension.

The advantages and disadvantages of hydrogels should be organized in a table format.

Examples of bioprinting should be illustrated (materials, cell types, results).

Author Response

Thank you for the opportunity to review the paper “Three-Dimensional Models of the Dental Pulp: Bridging Fundamental Biology and Regenerative Therapy,” which comprehensively explains current advances in three-dimensional culture models for dental pulp regeneration research. The structure of this paper is clear and concise, and it introduces the latest literature. The clarification of the relationship between dental pulp organoids and the tooth-on-a-chip model is particularly commendable. This article summarizes each type of 3D culture method and provides readers with a comprehensive explanation of the underlying science and clinical applications. On the other hand, there is redundant description and a lack of critical analysis, and a more thoughtful analysis is desired for a review article.

Major concern

The authors introduce the strengths and weaknesses of each model, but their critique of issues concerning experimental evidence and reproducibility is lacking.

We thank the reviewer for this important point. We have thoroughly addressed limitations related to experimental evidence and reproducibility throughout the manuscript (in sections 2.2, 2.3, 2.4, 2.6 and 4.4). These additions highlight current constraints and reinforce the importance of rigorous validation for future translational applications.

Section 4.6, “Bridging Translation to the Clinic” should be expanded to strengthen the description of practical challenges for regenerative endodontics.

We thank the reviewer for this suggestion. Section 4.6 has been expanded to provide a more detailed discussion of the practical and translational challenges associated with regenerative endodontic therapies, including regulatory considerations, manufacturing constraints, cell sourcing and standardization, delivery strategies, and the need for robust preclinical testing. A table has also been added to summarize the main differences between cell-free biologics, ATMPs, combined ATMP-device therapies, and implantable medical devices.

Minor concern

The expression “Collectively” is repeated in multiple places and should be revised.

We thank the reviewer for this observation. We have revised the manuscript to replace repeated instances of “Collectively” with more varied and context-appropriate expressions, ensuring clarity and improving readability throughout the text.

Figure 1 alone provides insufficient information.

Thank you for your comment. We have addressed this by adding additional figures and tables that provide more information to complement Figure 1.

Adding a conceptual diagram comparing each model (spheroid, organoid, tooth-on-a-chip) would enhance reader comprehension.

Thank you for the suggestion. We have added a conceptual diagram comparing models to enhance reader comprehension.

The advantages and disadvantages of hydrogels should be organized in a table format.

We thank the reviewer for this suggestion. The advantages and disadvantages of injectable hydrogels have been summarized in a table format in the revised manuscript for clarity.

Examples of bioprinting should be illustrated (materials, cell types, results).

We thank the reviewer for this suggestion. Specific examples of 3D bioprinting, including materials, cell types, and results, have been added in Section 2.3, 3D Bioprinted Constructs.

Round 2

Reviewer 1 Report

Comments and Suggestions for Authors

Accept in present form.